# Connecting Epigenetic and Genetic Diversity of LTR Retrotransposons in Sunflower (*Helianthus annuus* L.) and *Arabidopsis thaliana* L.

**DOI:** 10.3390/plants15020204

**Published:** 2026-01-09

**Authors:** Kirill Tiurin, Mikhail Kazancev, Pavel Merkulov, Yakov Demurin, Alexander Soloviev, Ilya Kirov

**Affiliations:** 1All-Russia Research Institute of Agricultural Biotechnology, Timiryazevskaya Str. 42, 127550 Moscow, Russia; tiurin.kn@gmail.com (K.T.); irelanddets@gmail.com (M.K.); paulmerkulov97@gmail.com (P.M.); a.soloviev70@gmail.com (A.S.); 2Moscow Center for Advanced Studies, Kulakova Str. 20, 123592 Moscow, Russia; 3Pustovoit All-Russia Research Institute of Oilseed Crops, Filatova St. 17, 350038 Krasnodar, Russia; yakdemurin@yandex.ru; 4All-Russia Center for Plant Quarantine, 140150 Ramenski, Russia

**Keywords:** nanopore sequencing, sunflower, *Arabidopsis*, LTR retrotransposons, DNA methylation profiling, epigenetics

## Abstract

Transposable elements (TEs) are ubiquitous components of plant genomes that profoundly influence plant diversity, adaptation, and genome structure. Transposition of TEs is primarily suppressed by distinct DNA methylation systems. However, the distribution of DNA methylation at the level of individual TEs in plants remains poorly understood. Here, we address this question by generating per-base cytosine methylation maps of individual long terminal repeat retrotransposons (LTR-RTEs) for the large sunflower (*Helianthus annuus* L.) and the small *Arabidopsis thaliana* genomes. *A. thaliana* was selected as the model species, for which genome-wide DNA methylation profiles have been extensively characterized in prior studies. Our analysis revealed significant heterogeneity in methylation patterns both between and within individual LTR-RTE lineages. We also found that the sunflower genes harboring intact or fragmented LTR-RTE insertions exhibit altered DNA methylation and expression profiles, with intact LTR-RTE insertions enriched in stress-response and regulatory pathways. Our interspecies comparison of DNA methylation patterns indicates that methylation patterns are intrinsic features of LTR-RTE lineages, conserved across diverse plant species but influenced by factors such as insertion age, element length, and proximity to genes. Furthermore, we identified epigenetically distinct clusters of *Tork* and *Athila* sunflower elements corresponding to separate phylogenetic clades, suggesting a link between epigenetic regulation and the genetic diversity of plant LTR-RTEs.

## 1. Introduction

Transposable elements (TEs) are DNA sequences capable of moving independently within the genome [1]. Barbara McClintock first discovered them in maize (*Zea mays*) [2]. It is now known that TEs are widely distributed across many organisms, including humans, where they constitute about 65% of the genome [3]. Plant genomes exhibit substantial diversity in TE content. For example, about ~21% of the *Arabidopsis thaliana* L. genome is occupied by transposons, whereas ~85% of the maize genome is composed of TE sequences [4,5]. The most widespread class of TEs in plants are long terminal repeat retrotransposons (LTR-RTEs), which transpose via a “copy-paste” mechanism using reverse transcription of RNA intermediates. LTR-RTEs significantly influenced plant genome evolution, plant breeding, and genetic diversity [3,6,7].

LTR-RTEs possess two identical LTR sequences that serve as promoters for the transcription of LTR-RTE genomic RNA and may contain various motifs for transcription factor binding [8]. Additionally, LTRs harbor transcription termination signals. Consequently, insertion of LTR-RTEs near or within genes can activate expression of downstream regions by providing 3′ LTRs or cause premature transcription termination [9]. Novel LTR-RTE insertions in exons do not always disrupt transcription or generate chimeric transcripts, as they can be efficiently spliced out during RNA processing [9,10]. Despite their important roles in plant biology and evolution, uncontrolled TE activity is generally deleterious. Given the risks posed by uncontrolled mobilome activity to plant fitness, LTR-RTEs are suppressed by multiple overlapping mechanisms, with small interfering RNAs, DNA methylation, and histone modifications playing major roles [11].

TE methylation in plants is a dynamic, context-dependent process that varies according to the genomic location and transcriptional activity of TEs. In *A. thaliana*, a species with a small genome, full-length and transcriptionally active TEs located in euchromatic regions are predominantly methylated by the RNA-dependent DNA methylation (RdDM) pathway, which establishes DNA methylation de novo through guidance by small RNAs [12]. Canonical RdDM involves the plant-specific RNA polymerases IV and V (Pol IV and Pol V) and primarily maintains methylation in regions that are already methylated [13]. Non-canonical RdDM initiates de novo methylation of new transposon insertions using 21–22 nucleotide siRNAs, resulting in sustained epigenetic repression [14,15]. It is assumed that novel LTR-RTE insertions are recognized by non-canonical RdDM, leading to cytosine methylation in all three sequence contexts, accompanied by DOMAINS REARRANGED METHYLTRANSFERASE 2 (DRM2) activity [4]. This represents the initiation stage of DNA methylation. Following this, the maintenance stage begins, during which RdDM-deposited CG and CHG methylation are replicated by DNA METHYLTRANSFERASE 1 (MET1) and CHROMOMETHYLASE 3 (CMT3), respectively, resulting in intense CG and CHG methylation across the LTR-RTE body. Concurrently, RdDM primarily targets CHH methylation at the edges of long LTR-RTEs, where short transcripts are generated by Pol IV and Pol V during the maintenance stage. This coordinated activity produces distinct DNA methylation patterns over LTR-RTE sequences, characterized by elevated CHH methylation at the LTRs and high CG/CHG methylation across the LTR-RTE body [16].

In contrast, heterochromatic TEs are maintained in a methylated state primarily via the activity of three DNA methyltransferases: MET1, CMT2 and CMT3 [17]. This DNA methylation maintenance is facilitated by the chromatin remodeler DECREASE IN DNA METHYLATION 1 (DDM1), which enables methyltransferase access to densely packed heterochromatic regions [12,18,19,20]. CHH methylation of transposons in heterochromatic regions remains relatively uniform across the entire length of the mobile elements [10].

Our understanding of DNA methylation distribution over transposable elements and the underlying molecular mechanisms has largely been derived from studies on *Arabidopsis thaliana* L., a model plant with a relatively small genome. However, multiple investigations in plants with large genomes, such as maize, have revealed substantial differences in both the distribution and maintenance of DNA methylation over LTR-RTEs. Comparative genome-wide bisulfite sequencing indicates that maize exhibits approximately twofold higher levels of CG and CHG methylation globally but roughly half the CHH methylation levels observed in *Arabidopsis* [21]. Importantly, CHH methylation in maize is predominantly localized to transposon–gene boundaries rather than throughout transposon bodies, underscoring the role of RdDM in restricting the spread of active chromatin from gene-rich euchromatic regions into adjacent TEs. This preserves gene expression domains [16]. In contrast, RdDM in *Arabidopsis* primarily functions to establish and maintain genome-wide transposon silencing. These divergent functional roles are largely attributable to fundamental differences in genome architecture: while transposons in *Arabidopsis* are largely confined to pericentromeric heterochromatic knob regions, maize features a more dispersed transposon landscape, with transposons extensively interspersed throughout chromosomes along with genes. Additionally, the notably low CHH methylation in maize compared to *Arabidopsis* is largely explained by maize’s lack of a functional CMT2 chromomethylase, which is crucial for heterochromatic CHH methylation maintenance in *Arabidopsis* [18]. Thus, methylation distribution patterns and functional dynamics may differ significantly between plants with small versus large genomes. While the former has been extensively studied over the past decade, DNA methylation mechanisms and patterns in large-genome plants remain comparatively undercharacterized. The identification of DNA methylation distribution patterns across repetitive sequences in large genomes is challenging using conventional bisulfite sequencing methods, commonly considered the “gold standard” for DNA methylation detection. However, long-read sequencing technologies such as PacBio and Oxford Nanopore have enabled both *de novo* genome assembly and precise identification of cytosine methylation within repetitive regions [22]. These novel platforms facilitate high-precision determination of DNA methylation at individual repeat elements, providing unprecedented insights into methylation patterns at the level of individual cytosines.

The sunflower (*Helianthus annuus* L.) has a large (~3.5 Gb) and complex genome, with a high prevalence of TEs [23]. Comparative genomics and cytogenetic studies indicated that nested insertions of TEs in sunflowers form large blocks of heterochromatin [24,25,26]. Furthermore, most of the sunflower TEs are young LTR-RTEs which potentially were active during *H. annuus* speciation [27,28,29].

In this study, we investigated the DNA methylation landscape of LTR-RTEs in the sunflower (*Helianthus annuus*) genome and integrated these methylation data with structural features and transcriptional activity. We found that sunflower LTR-RTEs exhibit pronounced inter- and intra-lineage variation in cytosine methylation (Cme) profiles across their full sequences in the CG, CHG, and CHH contexts. Furthermore, we showed that different LTR-RTE lineages encompass families with distinct and diverse epigenetic patterns. Collectively, our results provide a comprehensive methylome map of the sunflower genome and reveal substantial differences in methylation characteristics between plants with large and small genomes.

## 2. Results

### 2.1. Whole-Genome Nanopore Sequencing of Helianthus annuus L.

To profile methylation patterns across the sunflower genome, we performed whole-genome nanopore sequencing, generating 6,787,487 reads with an N50 of 12.8 kb. Following basecalling and read mapping to the XRQ v2.0 reference assembly [23], we achieved coverage exceeding 19.2×. The list of full-length LTR-RTEs with >10× coverage included more than 6762 of the 7314 annotated ones (>92%). The methylation level of each covered cytosine residue in the *H. annuus* reference genome revealed a CG mean methylation level (MML) of 92.72% across all chromosomes followed by CHG and CHH levels of 76.13% and 14.85%, respectively. With an estimated genome size of ~3.5 Gb, the MML of the *H. annuus* genome is similar to that in its two relatives: *Taraxacum koksaghyz* and *Tragopogon pratensis* L. (Asteraceae family) (85.2–98.0%, 68.4–75.0%, and 9.7–16.0% for CG, CHG and CHH, respectively) [30,31]. A close look at each individual pseudochromosome revealed two MML drops on chromosomes 7 and 13 compared to the other chromosomes, which corresponds to two large gaps in the sunflower genome assembly (Figure 1A).

Next, we analyzed gene and LTR-RTE density which was performed for intact as well as partial elements using domain-, structural-, and homology-based approaches. The annotation revealed that 88.24% of the sunflower genome assembly belongs to LTR-RTEs corroborating previous studies [27,32,33]. Gene distribution curves are smooth and inversely related to CG/CHG methylation patterns while LTR-RTE curves are more dramatically changing with strong local peaks at the heterochromatic centromeric and pericentromeric regions of chromosomes (Figure 1B). The similar patterns were previously described for other plant species [34,35].

### 2.2. Methylation of Various Sunflower LTR-RTE Lineages

Annotation of LTR-RTEs in the sunflower genome identified 7314 full-length elements. Of them, 4548 elements (62.18%) were classified as *Gypsy* lineages, dominated by *Retand* (1837; 25.12%), *Athila* (1742; 23.82%), and *Tekay* (893; 12.21%). The remaining 2766 elements (37.82%) belonged to nine *Copia* lineages, with the most abundant being *Ale* (807; 11.03%), *Ikeros* (599; 8.19%), and *TAR* (469; 6.41%).

To investigate methylation levels of intact LTR-RTEs in the sunflower genome we analyzed mean binned methylation by CG/CHG/CHH contexts. Since ONT methylation calling precision depends on coverage, we filtered out LTR-RTE sequences having <10× average coverage [36]. This results in 6762 (92.45%) elements for downstream analysis. Comparison of the Cme level between LTR-RTEs and the flanking regions showed that LTR-RTE bodies were hypermethylated in all three contexts, which is consistent with data from other plant species [4].

Then, to be able to compare Cme patterns across LTR-RTEs we established a Cme profile for each of the analyzed LTR-RTEs. The cluster analysis of methylation patterns across all three contexts showed considerable heterogeneity among sunflower LTR-RTEs lineages. LTR-RTE elements within *Ty1*/*Copia* lineages *TAR*, *SIRE*, *Ivana*, *Ikeros*, *Bianca*, *Angela*, *Ale*, and *Ty3*/*Gypsy* lineages including *Tekay* and *Retand* revealed significant differences between lineages. CG methylation profiles exhibited local decrease (*TAR*, *SIRE*, *Ivana*, *Ikeros*, *Bianca*, *Angela*, *Ale*, *Tekay* and *Retand* lineages) or dramatic reduction over the central part of LTR-RTEs between two LTRs (*SIRE* lineages). Meanwhile, CHG methylation patterns largely paralleled those observed for CG methylation. Notably, in the *SIRE* and *Retand* lineages, CHG methylation remained consistently high even in regions where CG methylation levels were substantially decreased, suggesting differential regulation of methylation contexts within these lineages. Additionally, CHH methylation showed frequent peaks of elevation throughout the LTR-RTE sequence for most lineages rather than confinement to the boundary regions alone. Several lineages, including *Retand*, *SIRE*, *Tekay*, and the *TAR*, exhibited weak or negligible increases in CHH methylation along their entire length (Appendix A).

Surprisingly, we found substantial differences between the methylation patterns among members of the *Athila* and *Tork* lineages. Clustering of *Athila* and *Tork* elements revealed three and two distinct clusters, respectively. *Tork* cluster 1 elements displayed a sharp drop in CHG methylation within the LTR, while CG and CHH methylation remained uniform across the sequences. *Tork* cluster 2 exhibited an asymmetric drop in CG methylation across the inter-LTR region, accompanied by elevated methylation at the LTR boundaries and stable CHG methylation levels throughout the sequence (Figure 2B,C).

Clusters of *Athila* elements differed in Cme distribution in all three contexts. CG and CHG methylation was uniform in cluster 1 and had local decreases/increases in cluster 3. Elements of cluster 2 displayed a pronounced drop in DNA methylation in the LTR regions in all three contexts. Notably, elements of all three clusters demonstrated asymmetrical distribution of Cme over the entire sequence with reduced Cme level near the right boundary (Figure 3B,C).

To explain the epigenetic diversity, we analyzed the phylogenetic relationships among the elements within the *Athila* and *Tork* lineages. We found that the epigenetically distinct cluster 2 of *Tork* elements, as well as clusters within the *Athila* lineage, form clearly separate phylogenetic clades (Figure 2A and Figure 3A, detailed phylogenetic trees: Appendix A and Appendix A, respectively). Thus, *Tork* phylogenetic clades B and C correspond to cluster 2 with decreased CG methylation level of the TE body while other elements within the family relate to cluster 1-defined methylation pattern (Figure 2A,B). In the same way *Athila* phylogenetic clade C corresponds to a cluster 2 with strong local decreases in all methylation contexts nearby TE edges while other phylogenetic clades exhibit mixed methylation patterns, represented by clusters 1 and 3 (Figure 3A,B). Thus, the differentiation in methylation patterns observed within *Tork* and *Athila* elements reflects their intra-lineage phylogenetic relationships.

Altogether, analysis of DNA methylation distribution along LTR retrotransposons revealed substantial epigenetic and genetic heterogeneity both between different lineages and among elements within the same lineage.

### 2.3. DNA Methylation Patterns of LTR-RTEs in Sunflower and Arabidopsis

We then asked whether the observed Cme patterns are conserved between the same LTR-RTE lineages of sunflower and *Arabidopsis*. For this, we also performed DNA methylation calling for *Arabidopsis thaliana* L. Col-0 genotype using ONT data from our previous study [10]. The genome-wide analysis indicated that mean methylation levels are highest in the CG context, averaging around 28.17%, followed by CHG methylation at approximately 10.76%, and CHH methylation rarely exceeding 3.95%. These values are lower than those obtained for sunflower genomes by 3.3-, 7.1-, and 3.8-fold for CG, CHG and CHH, respectively. The results demonstrate hypermethylation of the sunflower genome compared to the small *Arabidopsis* genome.

Then, we compared the methylation patterns of the same LTR-RTEs lineages between sunflower and *A. thaliana*. For this analysis, we chose two lineages, *Tork* and *Ale*, possessing sufficient members in both species. We identified two and three clusters based on Cme distribution in sunflower and *Arabidopsis*, respectively. The results of comparative analysis of Cme patterns revealed that the members of two *Tork* clusters (sunflower clusters 1 and 2 and *Arabidopsis* clusters 2 and 3) have atypical and pronounced reduction in inter-LTR methylation in both species (Appendix A). Additionally, one cluster of *Arabidopsis* (cluster 1) contained the elements with negligible methylation levels in all three contexts and the elements with such low Cme level were not present in the sunflower genome. Conversely, cluster analyses of the *Ale* lineage demonstrated similarities between species. Although *Ale* cluster 1 of *Arabidopsis* does not resemble any other clusters in sunflower (Figure 4). Furthermore, we found that *H. annuus Ale* cluster 1 is enriched in a single clade within a phylogenetic tree (Appendix A) while *A. thaliana* phylogenetic clades show no enrichment in a specific methylation pattern (Appendix A).

These results demonstrated highly similar DNA methylation distribution patterns across corresponding LTR-RTEs lineages in *Arabidopsis* and sunflower. However, *Arabidopsis* exhibited a higher number of elements with very low DNA methylation levels within each lineage.

### 2.4. The Relationship Between Methylation and Various LTR-RTE Features

We next sought to identify additional features distinguishing clusters of LTR-RTEs belonging to the *Athila* and *Tork* lineages. We compared Cme patterns between genic and non-genic insertions of these elements. This analysis showed that elements from *Tork* cluster 2 exhibited clear differences in Cme between genic and non-genic locations (Figure 5). A distinctive feature of *Tork* cluster 2, a pronounced decrease in CG methylation within the inter-LTR regions, was detected only in non-genic insertions, whereas elements inserted within genes displayed a Cme distribution pattern similar to that of *Tork* cluster 1 (Figure 5).

We next performed a statistical evaluation of additional features distinguishing LTR-RTE clusters. For each LTR-RTE, the following characteristics were analyzed: GC content (%), insertion time (Mya), element length, and distance to the nearest gene (bp). Statistical significance was assessed using the Wilcoxon rank-sum test. *Tork* elements from the two clusters differed significantly in insertion time (*p* < 0.0014), GC content within LTRs (*p* < 6.3 × 10^−5^), and length (*p* < 4.9 × 10^−14^). Elements from cluster 1 were generally longer, of intermediate age, and exhibited average GC content, whereas those from cluster 2 were shorter, more recently inserted, and had lower GC content.

A similar analysis of the *Athila* lineage showed that cluster 3 differed significantly from cluster 1 in all examined characteristics. Cluster 3 elements were more distant from genes, were longer on average, exhibited greater variability in local gene density, and included a higher proportion of elements with ancient insertions. In contrast, cluster 2 displayed the highest GC content across both the LTR and internal regions (Figure 6).

Together, these results suggest that the identified clusters of LTR-RTEs reflect a combination of insertion age, sequence composition, and genomic context, which likely act together to shape their Cme distribution patterns.

Despite the high heterogeneity, we were unable to reliably associate the significantly different parameters across clusters of different families with specific features of methylation patterns.

### 2.5. LTR-RTE Insertion Associated Changes in Gene Regulation

To assess the impact of LTR-RTEs insertions on Cme and transcriptional regulation of the genes with insertions, sunflower genes were classified into three categories: genes containing intact TE insertions (iTE; *n* = 307, 0.4%), genes with fragmented TE insertions (fTE; *n* = 10,171, 16.1%), and genes without LTR-RTE insertions (lTE; *n* = 52,791, 83.5%). Both iTE and fTE genes exhibited significantly higher levels of DNA methylation across all sequence contexts (CG, CHG, and CHH) compared with lTE genes (Appendix A; *p*-value < 0.05), suggesting that genic LTR-RTE insertions are associated with local hypermethylation.

Gene methylation (gM) in flowering plants typically occurs in two major patterns: (i) gene body methylation (gbM), characterized by CG methylation enrichment within the transcribed region and depletion of methylation at the transcription start (TSS) and termination (TTS) sites; and (ii) CHH/RdDM-type methylation, enriched for CG/CHG/CHH methylation across the gene body [37,38]. In sunflower, lTE genes displayed a gbM pattern that is frequently observed in angiosperms, whereas both fTE and iTE genes showed a CHH/RdDM-like profile (Figure 7). Notably, iTE genes exhibited elevated CG/CHG/CHH methylation outside of the gene body, with a local methylation depletion approximately 1 kb upstream and downstream of the TSS/TTS, respectively.

Expression levels differed substantially among the three gene groups. lTE genes had a greater proportion of expressed genes with a mean TPM 18.9 compared with fTE (0.3 TPM) and iTE (2.2 TPM) genes (Appendix A). This pattern mirrors observations in some species but differs from observations in maize where genes with and without insertions have similar levels of expression [21,39,40]. These results corroborate the higher Cme in the genes with insertions. Gene Ontology (GO) enrichment analysis revealed distinct biological associations among gene categories. lTE genes were enriched in housekeeping and core metabolic functions, including cellular organization (GO:0044237), regulation of metabolic processes (GO:0050794), and macromolecule metabolism (GO:0009058). fTE genes were enriched in signaling-related processes such as protein phosphorylation (GO:0016310, GO:0006468) and nucleic acid metabolism (GO:0006139). iTE genes showed enrichment in regulatory and stress-related functions, including phosphorus metabolism (GO:0006793, GO:0006796), transcriptional regulation (GO:0006357), DNA integration (GO:0015074), and defense responses (GO:0009617, GO:0042742). These results suggest that genic TE insertions are preferentially associated with genes involved in environmental response and regulatory flexibility (Appendix A).

To examine whether the position of LTR-RTE insertions within genes affects methylation, iTEs were classified as exonic, intronic, TSS-overlapping, or TTS-overlapping [41]. Among 307 intact insertions, 42 (13.7%) were exonic, 149 (48.5%) intronic, 37 (12.1%) overlapped TSS regions, 49 (16.0%) overlapped TTS regions, and 30 (9.8%) remained unclassified. We found that exonic and intronic LTR-RTE insertions are responsible for the U-shaped distribution of CG, CHG and CHH methylation in iTE genes. We analyzed LTR-RTE methylation relative to the genes containing the corresponding LTR-RTE inserts (Figure 8). Furthermore, we observed very high levels of CG and CHG methylation over the LTR-RTE sequence compared to the rest of the gene sequence. These results indicate that sunflower has sufficient mechanisms for specific recognition and methylation of LTR-RTEs within genes.

Then we wonder how LTR-RTE methylation affects gene expression. We found negative correlations for the insertion in TSS and TTS regions of the genes corroborating the important regulatory roles of these regions for transcription.

Beyond methylation effects, LTR-RTEs can directly influence transcription by contributing regulatory sequences. We identified three representative examples of sunflower genes producing chimeric transcripts, where transcription initiates within the LTR-RTE or includes LTR-RTE-derived exonic sequences: LOC110900634 (disease-resistance protein), LOC110919878 (LRR receptor-like serine/threonine-protein kinase), LOC110888714 (GAF-domain containing protein) (Figure 9A–C). Although these insertions are intronic, they disrupt normal transcription of the genes in two ways. Firstly, the LTR-RTEs are partially exonized (LOC110900634 and LOC110919878). Secondly, LTR-RTEs trigger premature termination of transcription (all three genes). We further analyzed Cme distribution over genes carrying insertions and a housekeeping gene (*HaRPB2*, LOC110937867, encoding RNA Pol II largest subunit) without insertions. Surprisingly, we did not find that the CG DNA methylation was specific to the LTR-RTE sequence but rather was observed over the whole length of the genes, including a housekeeping gene. However, CHG and CHH methylation was much higher for the genes carrying insertions compared to housekeeping genes. Moreover, we observed a substantial drop in Cme near TSS and TTS of the genes. For gene LOC110888714 we identified two such Cme sites and one of which co-localized with the LTR sequence of the inserted LTR-RTE. This site coincides with the observed alternative TTS for this gene, suggesting possible chromatin reorganization induced by LTR-RTE insertion.

## 3. Discussion

Previous studies have investigated DNA methylation differences at the level of LTR-RTE superfamilies and individual lineages [40,42,43], or compared methylation patterns in flanking regions between distinct insertions [44]. While these studies have revealed substantial differences in methylation distribution between phylogenetically distant LTR-RTE groups, they have not comprehensively addressed methylation heterogeneity within individual lineages at single-element resolution. Our study extends this knowledge by analyzing insertion-specific DNA methylation patterns, revealing that different LTR-RTE lineages exhibit characteristic methylation profiles at the lineage level, alongside intra-lineage heterogeneity, which is associated with evolutionary divergence within these lineages. Cross-species comparison between *Helianthus annuus* and *Arabidopsis thaliana* further supported this finding: the *Tork* and *Ale* lineages displayed remarkably similar patterns between both species consistent with previous reports demonstrating conservation of lineage-specific methylation signatures in *Glycine max* and *Vigna radiata* [40]. These findings establish that DNA methylation patterns are intrinsic characteristics of specific LTR-RTE lineages and phylogenetic clades in the lineages. This suggests that epigenetic silencing mechanisms operate through refined discrimination between different LTR-RTE families despite their evolutionary distance.

He et al. (2012) demonstrated that DNA methylation in *Arabidopsis* TEs depends on multiple factors: TE family identity, genomic location, insertion time, and element length [45]. Our analysis similarly identified insertion time, lineage identity, and element length as statistically significant variables (*p* < 0.05) distinguishing LTR-RTEs with distinct DNA methylation patterns. Notably, we found no significant differences in chromosomal location alone (pericentromeric vs. distal chromosomal regions), proposing that regional chromatin landmarks are insufficient predictors of methylation status in the sunflower genome. Instead, we identified proximity to genes as a critical feature correlating with methylation pattern variation. Specifically, elements of *Tork* and *Athila* lineages distinguished by Cme patterns showed significant differences in their distance to the nearest genes (*p* < 0.05). This observation aligns with findings in maize, where DNA methylation functions primarily to protect genes from deleterious TE-mediated transcriptional disruption, rather than serving as the primary TE silencing mechanism [21]. This pattern, where methylation intensity is determined by genomic microenvironment and proximity to genes may represent a characteristic feature of large plant genomes.

Contrasting with maize, where methylation in all three contexts (CG, CHG, CHH) is frequently concentrated at LTR-RTE edges [44], we did not observe such a picture for any sunflower LTR-RTEs. Instead, *SIRE* and *Tork* lineages in non-genic locations exhibit distinct CG methylation peaks at element boundaries, while CHG and CHH methylation remains relatively uniform throughout the sequences. This position-dependent pattern was strongly associated with non-genic insertions, whereas genic insertions displayed more diffuse methylation profiles. These observations suggest that sunflower has evolved a boundary-protection mechanism to prevent methylation spreading from TE sequences into adjacent genes, similar to the system characterized in maize [21,46]. This boundary-protection model suggests that host cell methylation machinery exhibits spatial specificity: methylation is preferentially maintained at TE edges, where it can suppress TE transcription without compromising gene expression rather than distributed uniformly across the TE or spreading into flanking chromosomal regions. The fact that this pattern is position-dependent (present in non-genic insertions but disrupted in genic insertions) suggests that local chromatin organization and transcriptional architecture of flanking genes may modulate methylation deposition patterns.

The considerable phylogenetic divergence between LTR-RTE superfamilies and lineages presents significant challenges for investigating the relationship between their genetic and epigenetic variability. In this study, we leveraged advances in long-read nanopore sequencing to perform LTR-RTE insertion-specific DNA methylation profiling at individual cytosine contexts (CG, CHG, and CHH) across the large sunflower genome. A striking finding of this study is that most LTR-RTE lineages display conserved methylation patterns among their members, yet two lineages, *Tork* and *Athila*, exhibit substantial and consistent intra-lineage heterogeneity. Cluster analysis revealed that *Tork* elements segregate into two distinct methylation clusters: cluster 1 elements were significantly longer (*p* < 4.9 × 10^−14^), of intermediate insertion age, and displayed uniform methylation across all contexts, whereas cluster 2 elements were shorter (*p* < 4.9 × 10^−14^), more recently inserted (*p* < 0.0014), and exhibited pronounced methylation valleys in internal regions. Similarly, *Arabidopsis Tork* and *Ale* lineages exhibited unexpected methylation heterogeneity, indicating this phenomenon is not species-specific.

Previous studies proposed that element length determines methylation patterns through a PolIV/V-dependent mechanism: longer TEs should experience higher transcription by RNA Polymerase IV/V, leading to increased siRNA production that directs RdDM to specific TE loci [43]. While element length emerged as the most discriminating feature distinguishing Cme-based clusters within *Tork* lineages, this model alone inadequately explains the observed heterogeneity. Collectively, our observations indicate that intra-lineage methylation heterogeneity reflects complex interactions between multiple variables: insertion time, sequence composition (GC content, structural motifs), element length, and genomic context. The relative contribution of each factor likely varies between lineages, explaining why some families (*Tork*, *Athila*) exhibit substantial heterogeneity while others (*Ale*, *Retand*) maintain relatively uniform methylation patterns.

The relationship between transposable element sequences and DNA methylation is not unidirectional. Rather, some TE lineages encode cis-acting recognition motifs and trans-acting protein factors that actively shape their methylation distribution patterns. A compelling example is the *VANDAL* family of DNA transposons in *A. thaliana*, where individual members encode sequence-specific anti-silencing proteins (VANC proteins) that bind to tandem repeat motifs within *VANDAL* non-coding regions and induce demethylation of their cognate elements [47,48]. VANC21 specifically recognizes nine-base target motifs within *VANDAL21* tandem repeats, and methylation loss radiates from these binding sites across both CG and non-CG contexts, affecting the entire element length. This anti-silencing system reflects an ongoing evolutionary arms race: while host RNA-directed DNA methylation machinery targets non-coding regions of active *VANDAL* elements to re-silence them, rapid diversification of VANC target sequences allows different *VANDAL* families to evolve distinct and non-overlapping anti-silencing specificities [49]. We previously demonstrated that transpositionally active LTR-RTE members of the sunflower *Tekay* lineage encode an additional open reading frame (ORF) that produces a protein with structural homology to dsRNA-binding proteins [32]. Whether similar ORFs are encoded by *Tork* and *Athila* lineages and whether such encoded proteins modulate the epigenetic landscape of these elements remains to be investigated. These findings suggest that the LTR-RTE sequence heterogeneity we observed in this study at the intra-lineage level may result not solely from host-directed factors, but from LTR-RTE-encoded factors that actively modulate methylation patterns. This model, where individual LTR-RTE elements are active participants in their epigenetic regulation rather than passive targets of host silencing machinery, provides an alternative explanation for why phylogenetically related LTR-RTE elements display dramatically different methylation profiles despite sequence homology. This finding also provides a foundation for understanding the mechanisms controlling this variation and may inform future development of novel TE-based mutagenesis strategies and epigenetic diversification approaches for crop improvement.

Our results demonstrate that LTR-RTE insertions in the sunflower genome are closely associated with local epigenetic and transcriptional remodeling. Genes containing intact or fragmented LTR-RTE insertions exhibit strong hypermethylation across all contexts and shift from the canonical gene body methylation (gbM) to a CHH/RdDM-type methylation pattern. This shift likely reflects the recruitment of small interfering RNAs (siRNAs) and activation of the RNA-directed DNA methylation (RdDM) pathway targeting TE sequences, resulting in methylation that can extend into adjacent regions. This observation parallels the co-spreading of siRNA and methylation reported in *A. thaliana* [50]. However, in *A. thaliana* and *Brachypodium distachyon*, which possess compact genomes with high gene density, TE-induced methylation spreading is limited [10,51]. In contrast, *O. sativa* and *Z. mays* exhibit methylation spreading that depends on TE age, recombination rate, and insertion context [44,52,53]. In maize, methylation spreading is also associated with distinct histone modifications, including constitutive heterochromatic H3K9me2 for spreading TEs and facultative heterochromatic H3K27me3 for non-spreading TEs. This suggests that the transition from intact to fragmented elements may be linked to chromatin relaxation [44]. Functionally, sunflower genes carrying LTR-RTE insertions tend to be less expressed, consistent with previous studies showing that TE insertions often repress nearby transcription. However, the enrichment of LTR-RTE-associated genes in stress- and regulatory-related GO terms suggests that some LTR-RTE insertions may confer adaptive regulatory flexibility, potentially by supplying novel regulatory motifs or promoters responsive to environmental cues. Finally, the identification of gene-TE chimeric transcripts illustrates the potential of intact LTR-RTEs to participate directly in transcriptional processes.

## 4. Materials and Methods

### 4.1. Plant Material

Sunflower seeds of the “ZS” line were provided by the Pustovoit All-Russia Research Institute of Oilseed Crops (Krasnodar, Russia). Seeds were soaked overnight in Petri dishes with distilled water, and then planted in a mixture of peat and perlite (3:1 ratio).

### 4.2. DNA Isolation

For DNA isolation, plant tissues were homogenized in liquid nitrogen to a fine powder state, and 100 mg was used for DNA isolation using a CTAB-based protocol [54].

### 4.3. RNA Isolation

Total RNA was extracted from the second pair of true leaves, using 100 mg of tissue per technical replicate. Material was homogenized in liquid nitrogen.

To the homogenized material, 1 mL of ExtractRNA (Evrogen, Moscow, Russia) was immediately added and homogenized until the tissue thaws, then transferred to a 1.5 mL tube. The tubes were incubated at room temperature for 15 min.

200 µL of chloroform was added to the solution, and it was incubated at room temperature with gentle flipping for 5 min. The mixture was centrifuged for 10 min at 13,500 rcf, 4 °C. Without disturbing the interphase, the supernatant was transferred to new sterile tubes, 500 µL of chloroform was added. The mixture was centrifuged again for 10 min at 13,500 rcf, 4 °C. Without disturbing the interphase, the supernatant was transferred to new sterile tubes, 50 µL of 3M AcONa (pH 5.2) and 500 µL of isopropanol were added, and the mixture was incubated for 1 h at −20 °C or 15 min at −80 °C. It was centrifuged for 10 min at 13,500 rcf, 4 °C. Without disturbing the pellet, the supernatant was removed, then 1 mL of freshly prepared 75% ethanol was added. The mixture was centrifuged for 10 min at 13,500 rcf, 4 °C. Carefully, without disturbing the pellet, the supernatant was removed. The pellet was air-dried and then dissolved in 50 µL of nuclease-free water.

After extraction, RNA was re-precipitated to remove guanidinium thiocyanate. We brought the volume of the sample up to 400 µL and 300 µL of chloroform was added. The mixture was centrifuged for 15 min at 13,500 rcf, 4 °C. Carefully, without touching the interphase, the supernatant was transferred to new sterile tubes. Then, 400 µL of isopropanol and 40 µL of 3 M AcONa (pH 5.2) were added. The mixture was centrifuged for 15 min at 13,500 rcf, 4 °C. The supernatant was removed, and 1 mL of 80% ethanol was added. It was centrifuged for 5 min at 13,500 rcf, 4 °C. Carefully, without disturbing the pellet, the supernatant was removed. The pellet was air-dried and then dissolved in 40 µL of nuclease-free water.

The RNA concentration and the quality of the isolated RNA were estimated by nanodrop (Nanodrop Technologies, Wilmington, CA, USA) and gel electrophoresis on a 1.2% agarose gel with ethidium bromide staining. Results were visualized using a Gel Doc XR+ UV camera (Bio-Rad, Hercules, CA, USA).

### 4.4. Synthesis of Double-Stranded Complementary DNA (ds-cDNA)

A total of 2 µg RNA (without DNase-treatment) was used for reverse transcription. Reverse transcription was carried out using the MINT cDNA synthesis kit (Evrogen, Moscow, Russia) following the manufacturer’s instructions. First-strand synthesis was initiated with poly(A)-specific primers. The optimal number of PCR cycles (17 cycles) was adapted to reach the exponential phase of amplification. The ds-cDNA concentration and quality were assessed using NanoDrop (NanoDrop Technologies, Wilmington, CA, USA) and Qubit (Qubit dsDNA BR Assay Kits, Thermo Fisher Scientific, Waltham, MA, USA) and gel electrophoresis. Synthesized ds-cDNA was purified by 1.8× Agencourt AMPure XP Beads (Beckman Coulter, Pasadena, CA, USA) in accordance with the manufacturer’s instructions.

### 4.5. Nanopore Library Preparation and Sequencing

Prior to library preparation, total DNA from sunflower was end-repaired using the NEBNext Ultra II End Repair Module (New England Biolabs, Ipswich, MA, USA). For whole-genome sequencing, libraries were prepared using the Ligation Sequencing Kit 114 (SQK-LSK114, Oxford Nanopore Technologies, Oxford, UK).

For cDNA libraries, 400 ng of each double-stranded cDNA sample was phosphorylated in a 50 µL reaction containing 1 mM dATP, 10× T4 polynucleotide kinase buffer, and 10 U of T4 polynucleotide kinase (Sibenzyme, Novosibirsk, Russia). Phosphorylated samples were purified using a 1:1 ratio of AMPure XP beads (Beckman Coulter, Brea, CA, USA) according to the manufacturer’s instructions. The purified ds-cDNA was used for library preparation with the Native Barcoding Kit 96 V14 (SQK-NBD114.96, Oxford Nanopore Technologies, Oxford, UK).

### 4.6. Reference Genomes

In the current study, for *H. annuus* HanXRQr2.0-SUNRISE (RefSeq accession: GCA_002127325.2) genome assembly and original genome annotation was used [23], for *A. thaliana* TAIR10.1 (RefSeq accession: GCF_000001735.4) genome assembly and Araport11 gene annotation was used [55].

### 4.7. Transposable Elements’ Annotation

For both *H. annuus* and *A. thaliana* reference genomes Long Terminal Repeat Retrotransposons (LTR-RTEs) were annotated using DANTE v.0.2.6 with the following options: “-D Viridiplantae_v4.0 -M BL80” and DANTE_LTR v.0.4.0.5 with the following options: “-M 0 --no_ambiguous_domains” [56]. For whole-genome annotation of LTR-RTEs in the *H. annuus* reference genome, sequences of obtained intact LTR-RTEs were extracted from the reference genome with respect to strand using bedtools getfasta v2.27.1 (“-s” option) and used as a custom library for RepeatMasker2 v.4.1.5 (Smit, AFA, Hubley, R. & Green, P “RepeatMasker” at (http://www.repeatmasker.org, last accessed on 19 November 2025) (search engine rmblastn v.2.14.0, “-e rmblast -qq -norna -no_is” options) [57,58]. The analysis was limited to LTR-RTEs because the content of the other TE sequences (that is, LINEs, SINEs, and DNA-TEs) in the sunflower genome was negligible [27,59,60].

### 4.8. Methylation Calling

Raw R10 pod5 data were basecalled using Dorado v.0.7.2 https://github.com/nanoporetech/dorado (last accessed on 19 November 2025) with the model r10.4.1_e8.2_400bps_sup@v.5.0.0. Output unaligned BAM files were mapped to the reference genome using minimap2 v2.17 with the following options: “-ax map-ont” [61]. Supplementary, secondary alignments and unmapped reads were filtered out using samtools view v.1.9 (“-F 0x904” option), filtered BAM files were sorted and indexed using samtools and visualized using JBrowse2 v.3.6.5 [62,63].

Consensus cytosine methylation was called from obtained BAM files using modkit pileup v.0.5.1 (https://github.com/nanoporetech/modkit, last accessed on 19 November 2025) with the following options: “--filter-threshold C:0.7 --ignore h” for *H. annuus* and “--filter-threshold C:0.75 --ignore h” for *A. thaliana*. Calls at positions with at least three reads were retained, and the remaining calls were split by cytosine context (CG, CHG, and CHH) using modkit motif bed and bedtools intersect.

Coverage of *H. annuus* intact LTR-RTE annotations (“transposable_element” feature, DANTE_LTR initial annotation) were calculated using python3 v.3.8.10 pysam v.0.23.3 library.

### 4.9. Methylation Analysis

To assess methylation level of an individual intact LTR-RTEs the average level of methylation at each context was calculated from passed filtering methylation calls in 40 bp bins for 2 kbp. flanks and in 50 bins for TE bodies. Methylation from TE bodies were clustered with pdist and hierarchy modules from skipy v.1.10.1 python3 package using correlation as a distance metric and average linkage method for sunflower calls and euclidean as a distance metric and ward as linkage method for *Arabidopsis* calls. The optimal number of clusters was assessed using the Elbow method.

Whole-genome distribution of methylation calls for *H. annuus* XRQr2.0 assembly was calculated in 100 kbp. bins. Average level of methylation at CG/CHG/CHH for both *H. annuus* and *A. thaliana* reference genomes were calculated as the mean of all methylation calls at each context.

### 4.10. Phylogenetic Analysis

The reverse transcriptase (RT) domain was used to build phylogenetic trees due to its high conservation in LTR-RTEs [64]. *Amino acid* sequences of RTs were retrieved from initial DANTE annotation obtained in the previous step and aligned using mafft v7.453 with default parameters [65]. Phylogenetic trees were constructed from obtained alignment using the neighbor-joining (NJ) method with IQ-TREE v.3.0.1 (https://doi.org/10.32942/X2P62N), with 1000 bootstrap replicates.

### 4.11. RNAseq Analysis and Chimeric Gene-TE Transcript Discovery

Raw ONT RNAseq reads were trimmed using Porechop v.0.2.4 (https://github.com/rrwick/Porechop (accessed on 20 September 2025)) (parameters: “--require_two_barcodes”). Then, trimmed reads were mapped to the HanXRQr2.0-SUNRISE reference genome using minimap2 with the following parameters: “-a -x splice”, obtained SAM files were converted to BAM file, filtered (only primary alignments were taken for further analysis), sorted, and indexed using samtools. Transcripts were collapsed using StringTie2 v.2.2.1 with “-f 0.1 -L -j 3 -t” parameters [66]. Transcripts from two biological replications were merged using the StringTie2 “--merge” option. Filtered and sorted BAM files as well as StringTie2 annotations were visualized in JBrowse2 v.1.6.5 for manual data curation [63].

For gene expression analysis, reads from two biological replications were counted using featureCounts v.2.0.0 and original HanXRQr2.0-SUNRISE gene annotation or obtained on the previous step LTR-RTE annotation with “-L --fracOverlap 0.5 --minOverlap 50 --largestOverlap -M --fraction” options [67]. Counts per gene were normalized using TPM units with bioinfokit v.2.1.4 *python* package (last accessed on 19 November 2025) [68].

Chimeric gene-TE transcripts were identified by following algorithm: (i) gene (‘exon’ feature type) and LTR-RTE annotations (‘transposable_element’ feature type, DANTE_LTR initial annotation) were separately intersected with StringTie2 assembled transcripts using bedtools intersect; (ii) if transcript is intersected with both gene and LTR-RTE annotation it was considered as chimeric and verified manually.

### 4.12. LTR-RTE Parameters Calculation

For each intact LTR-RTE number of parameters were estimated. Insertion time was calculated using the Kimura 2-parameter method [69], in brief, LTR pairs were aligned using mafft to obtain number of transitions and transversions [65], then insertion time was calculated by the following formula: T = K/2r, where T is the insertion time in Mya, K is the genetic distance, and r is the base mutation rate per year, which was taken as r = 1.0 × 10^−8^ [70]. GC-content percentages were calculated as sum of cytosine and guanine bases count per element divided on its length. The distance to the nearest gene was estimated as the minimal distance between LTR-RTE edge and any annotated gene boundary, irrespective of the strand orientation.

### 4.13. Gene Ontology Analysis of Genes Harboring Intact LTR-RTE Insertions

Gene ontology (GO) enrichment analysis for genes with intact LTR-RTE insertions was conducted using topGO v.2.26.0 R package and gene ontology file generated by NCBI for sunflower XRQr2.0 assembly. The analysis utilized Fisher’s exact test to calculate GO term enrichment *p*-values, significant enrichment was determined using 0.05 *p*-value cutoff.

### 4.14. Visualization

Methylation heatmaps, metaplots as well as other plots were visualized using R v.4.4.2 ggplot2 v.3.5.1, *patchwork* v.1.3.0 and ggpubr v.0.6.0 packages. Color palette was applied to each plot using R RColorBrewer v.1.1-3 package color schemes. Correlation coefficients were calculated using the *cor* function from the base R stats package. Wilcoxon rank-sum test was calculated using the wilcox.test function from the base R.

### 4.15. Code Availability

Code for methylation data analysis as well as R code for figure generation can be found at https://github.com/soyboy-hub/SUN_METHYLOME_paper (accessed on 1 September 2025).

## 5. Conclusions

Based on comprehensive insertion-specific analysis of DNA methylation patterns across LTR retrotransposons in the sunflower and Arabidopsis genomes, we reveal both lineage-level conservation and substantial intra-lineage heterogeneity in methylation profiles. Our results demonstrate that methylation patterns are intrinsic features of LTR-RTE lineages, conserved across divergent plant species, yet influenced by complex factors including insertion time, element length, and proximity to genes. Finally, we show that epigenetically distinct clusters of *Tork* and *Athila* elements correspond to phylogenetically distinct clades, suggesting a connection between epigenetic and genetic diversity in sunflower LTR retrotransposons.

## Figures and Tables

**Figure 1 plants-15-00204-f001:**
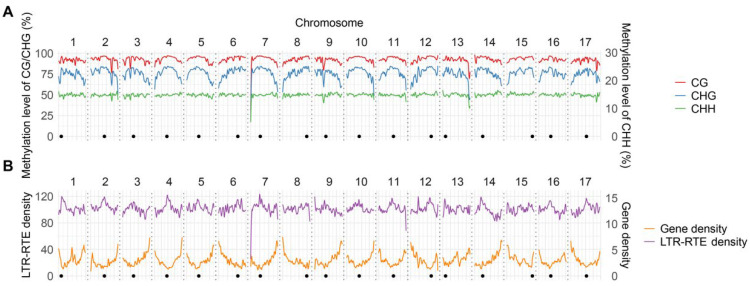
(**A**) Methylation pattern at CG, CHG and CHH context and (**B**) density of genes and LTR-RTEs across all chromosomes per 100 kb. Metaplots show smoothed mean (solid line) methylation level per bin, positions of the centromeres are plotted as points at the bottom of each subplot, the dotted lines define the start and end of each chromosome.

**Figure 2 plants-15-00204-f002:**
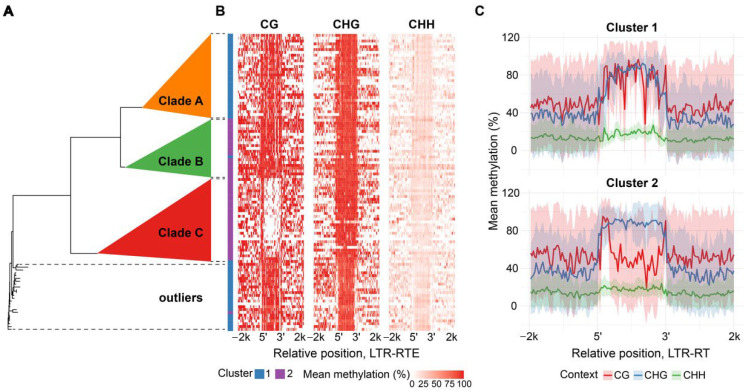
Methylation patterns and phylogeny of *H. annuus Tork* LTR-RTE lineage. (**A**) Phylogenetic tree of reverse transcriptases from individual elements, constructed using the neighbor-joining method. (**B**) Heatplot of LTR-RTEs average methylation level across element length for CG, CHG and CHH contexts, rows represent individual elements grouped on clusters, where each row corresponds to a leaf on the phylogenetic tree. (**C**) Methylation pattern of individual clusters which show smoothed mean (solid line) and standard deviation (shaded area) of binned methylation.

**Figure 3 plants-15-00204-f003:**
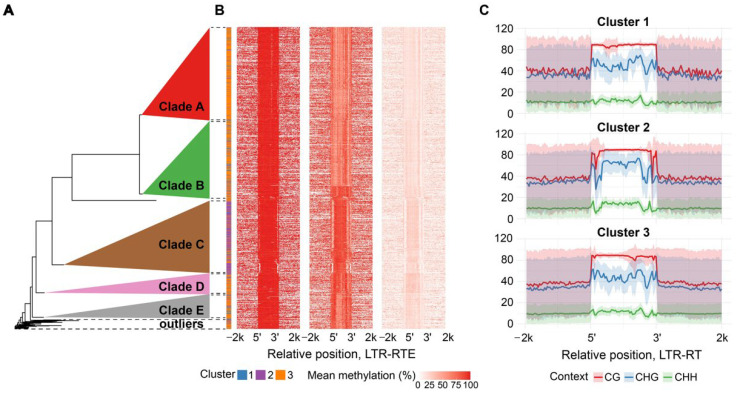
Methylation patterns and phylogeny of the *H. annuus Athila* LTR-RTE lineage. (**A**) Phylogenetic tree of reverse transcriptases from individual elements, constructed using the neighbor-joining method. (**B**) Heatplot of LTR-RTEs’ average methylation level across element length for CG, CHG and CHH contexts, rows represent individual elements grouped on clusters, where each row corresponds to a leaf on the phylogenetic tree. (**C**) Methylation pattern of individual clusters which show smoothed mean (solid line) and standard deviation (shaded area) of binned methylation.

**Figure 4 plants-15-00204-f004:**
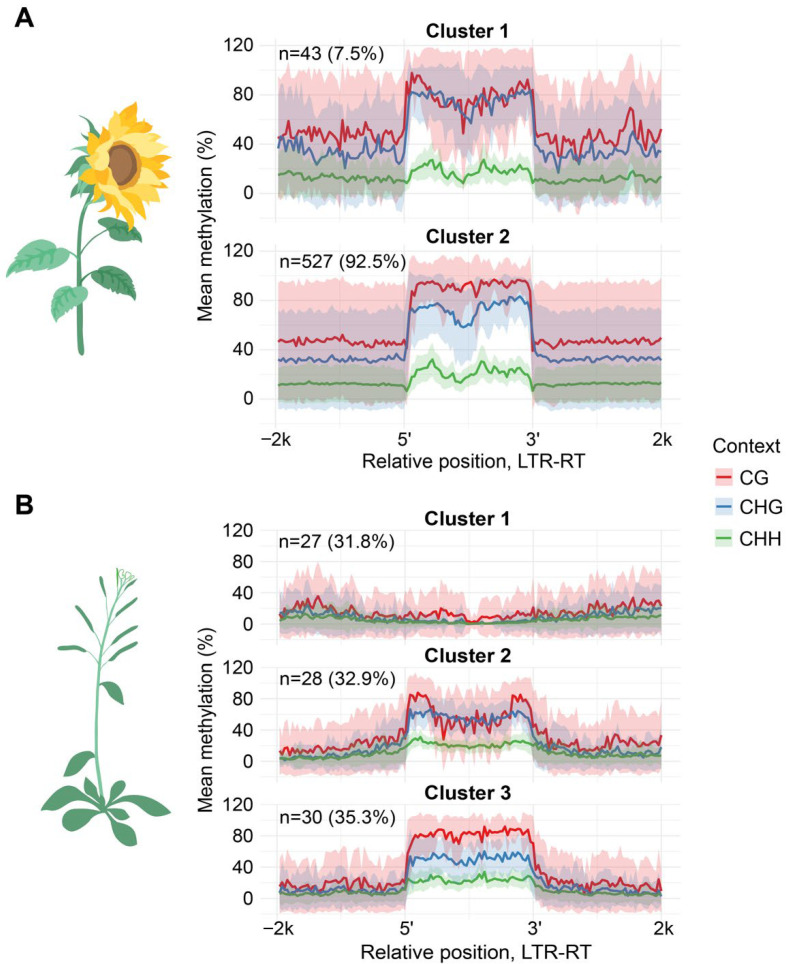
Methylation patterns of the *Ale* LTR-RTE lineage in (**A**) *H. annuus* and (**B**) *Arabidopsis thaliana*. Metaplots display smoothed mean methylation levels (solid line) and standard deviation (shaded area) across binned intervals within defined clusters.

**Figure 5 plants-15-00204-f005:**
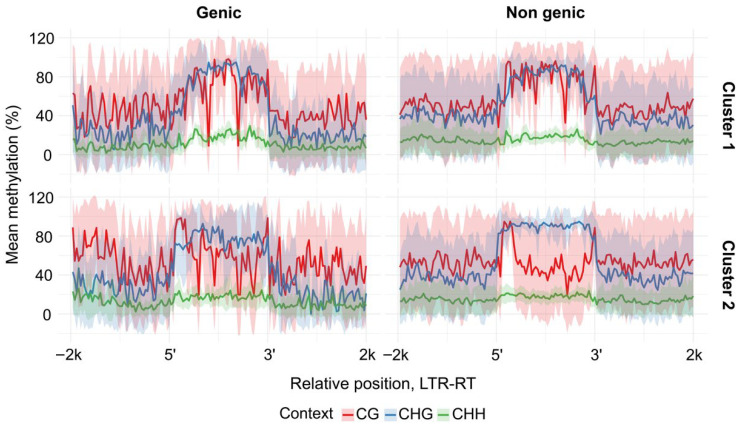
Methylation patterns across intergenic and intragenic *H. annuus Tork* LTR-RTEs. Metaplots display the smoothed mean methylation level (solid line) and standard deviation (shaded area) across binned genomic positions for defined element clusters.

**Figure 6 plants-15-00204-f006:**
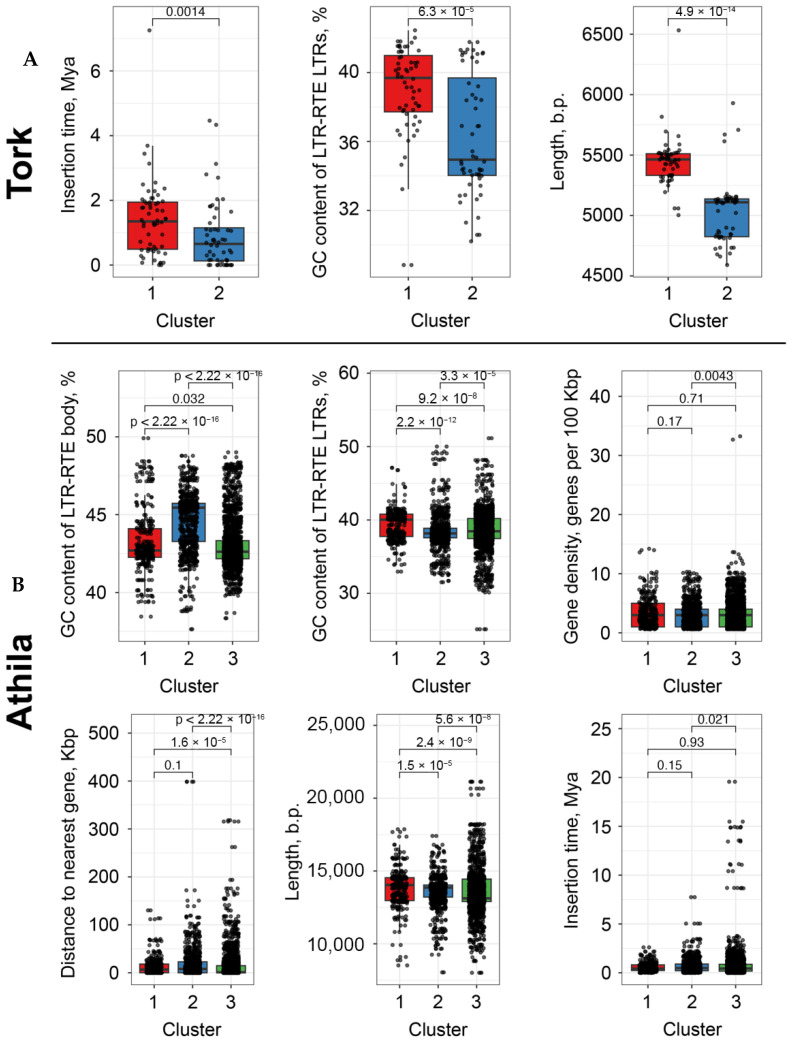
Statistical evaluation of parameters for (**A**) *Tork* and (**B**) *Athila* elements, including insertion time, CG content, length, local gene density and distance to the nearest gene. Statistical significance (Wilcoxon rank-sum test) between clusters is indicated as numerical value above boxplots.

**Figure 7 plants-15-00204-f007:**
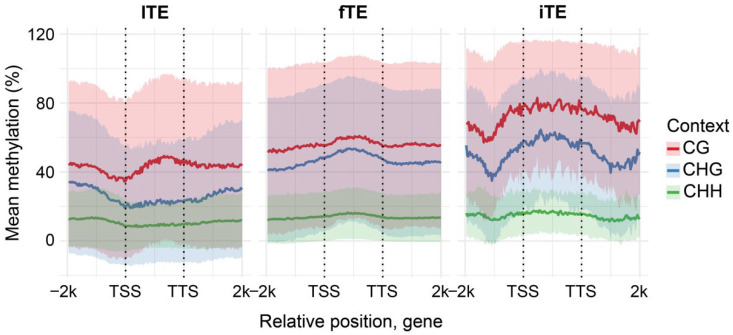
Gene methylation pattern across genes harboring fragmented TE insertions (fTE), intact TE insertions (iTE) and genes lacking TE insertions (lTE). Metaplots which show smoothed mean (solid line) and standard deviation (shaded area) of binned methylation across defined gene types.

**Figure 8 plants-15-00204-f008:**
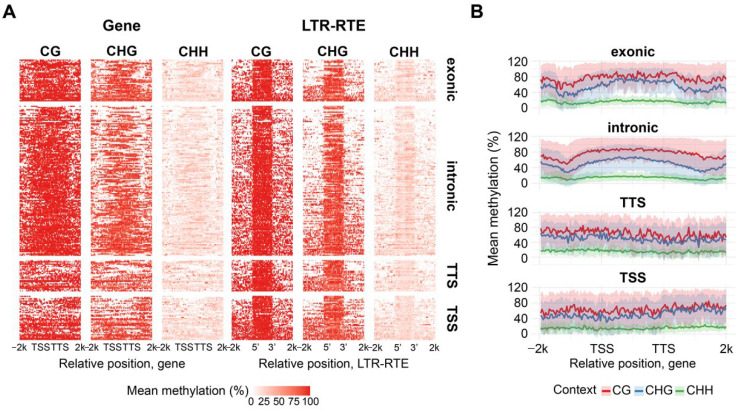
Methylation patterns of *H. annuus* genes harboring intact LTR-RTE insertions. (**A**) Heatplot of genes and LTR-RTEs average methylation level across gene or element length for CG, CHG and CHH contexts, rows represent gene–LTR-RTE pairs. (**B**) Metaplots which show smoothed mean (solid line) and standard deviation (shaded area) of binned methylation across genes harboring defined LTR-RTE insertion types.

**Figure 9 plants-15-00204-f009:**
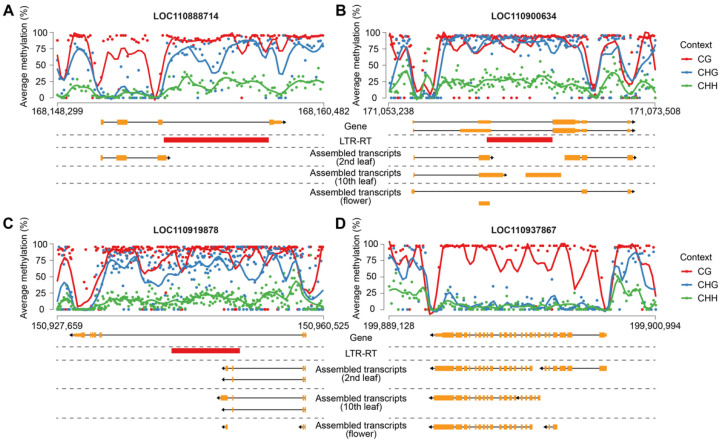
Methylation patterns in the third leaf along with gene annotation and transcript isoforms across three studied tissues. Metaplots show the smoothed mean (solid line) of binned methylation across genes; dots represent methylation level per bin. Orange rectangles represent exons (thick) or untranslated regions (UTRs, thin), lines represent introns, and arrows indicate the transcription direction. TEs are shown as red rectangles. (**A**) LOC11088714 gene; (**B**) LOC110900634; (**C**) LOC110919878, (**D**) LOC110937867.

## Data Availability

The nanopore whole-genome sequencing (WGS) and RNA-seq data for *Helianthus annuus* L. generated in this study are available in the NCBI Sequence Read Archive (SRA) under BioProject accession PRJNA1394510. Data for *Arabidopsis thaliana* L. (wild-type Col-0 ecotype) is available in the BioProject accession PRJNA1263850.

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
