# Peer review of "Connecting Epigenetic and Genetic Diversity of LTR Retrotransposons in Sunflower (Helianthus annuus L.) and Arabidopsis thaliana L."

_plants, 2026, doi:10.3390/plants15020204_

Round 1
Reviewer 1 Report
Comments and Suggestions for Authors
- Lines 18-19. In the Summary it appears that in the search for LTR-RTEs sequences, the whole process (from DNA extraction onwards) was carried out on both species arabidopsis and sunflower, while another situation is mentioned in Materials and Methods. Please make it clear in the Abstract that the whole process was carried out only on sunflower and that the data for arabidopsis were taken from previous studies.
- Lines 113-127. It is suggested to refocus the content of the last paragraph of the Introduction section. The way it is presented appears to be a great mixture of the sections on Materials and Methods, Results, and even Conclusions. Please focus the wording to clearly and concisely indicate the objectives of the study.
- Lines 676-677. I suggest modifying the first sentence of the Conclusions, as it resembles to Materials and Methods. I suggest replacing “We carried out a comprehensive, insertion-specific analysis of DNA methylation patterns across LTR retrotransposons …..” by “Based on comprehensive insertion-specific analysis of DNA methylation patterns across LTR retrotransposons …..”
- Typos: a) Throughout the document, the abbreviation μl is used to refer to microliters. b) Please be aware that the correct notation is μL.
- Line 140. please write (Asteraceae family) without italics.
Author Response
We thank the reviewer for the comments. Please find our point-by-point answers below. We believe these revisions have elevated the manuscript’s clarity and impact.
Q1. Lines 18-19. In the Summary it appears that in the search for LTR-RTEs sequences, the whole process (from DNA extraction onwards) was carried out on both species arabidopsis and sunflower, while another situation is mentioned in Materials and Methods. Please make it clear in the Abstract that the whole process was carried out only on sunflower and that the data for arabidopsis were taken from previous studies.
A1. We agree. All experimental procedures and genome sequencing was performed solely for sunflower, while data for Arabidopsis were obtained from our previous study (as detailed in the Material and Methods section). We corrected the abstract accordingly.
Q2. Lines 113-127. It is suggested to refocus the content of the last paragraph of the Introduction section. The way it is presented appears to be a great mixture of the sections on Materials and Methods, Results, and even Conclusions. Please focus the wording to clearly and concisely indicate the objectives of the study.
A2. The last paragraph of the introduction section has been modified.
Q3. Lines 676-677. I suggest modifying the first sentence of the Conclusions, as it resembles to Materials and Methods. I suggest replacing “We carried out a comprehensive, insertion-specific analysis of DNA methylation patterns across LTR retrotransposons …..” by “Based on comprehensive insertion-specific analysis of DNA methylation patterns across LTR retrotransposons …..”
A3. The Conclusion has been corrected.
Q4. Typos. a) Throughout the document, the abbreviation μl is used to refer to microliters. b) Please be aware that the correct notation is μL.
A4. Corrected.
Q5. Line 140. please write (Asteraceae family) without italics.
A5. Corrected.
Reviewer 2 Report
Comments and Suggestions for Authors
Major:
1) This manuscript has big problems with drawings – small font, low quality. It is difficult to understand what is presented in the figures. Thus, Fig. 1: authors should increase the font size. Fig. 2 – also increase the font size, probably present Fig. 2a as top separate figure, and 2b and 2c below 2a. The same for other figures.
2) The authors should explain why it was necessary to study LTR retrotransposons in two different plant species. What important information did the authors learn by using these two species together?
3) Why LTR retrotransposon and not other transposons?
4) Authors should include the Table with basic data of the nanopore sequencing for for each sample (total reads, reads after cleaning, GenBank accession numbers, etc).
5) Authors should improve all legends for figures and tables – explain used abbreviations, samples.
Minor:
7) Line 523: “material, 1 ml of (Evrogen, Moscow, Russia) was” - 1 ml of what?
8) Line 552: “4.4. Synthesis of ds-cDNA” – explain ds-cDNA abbreviation.
Comments on the Quality of English LanguageA small edit is needed
Author Response
We thank the reviewer for the comments. Please find our point-by-point answers below. We believe these revisions have elevated the manuscript’s clarity and impact.
Q1. This manuscript has big problems with drawings – small font, low quality. It is difficult to understand what is presented in the figures. Thus, Fig. 1. authors should increase the font size. Fig. 2 – also increase the font size, probably present Fig. 2a as top separate figure, and 2b and 2c below 2a. The same for other figures.
A1. We fully agree with this comment. We have increased font sizes across all figures (axes text, labels, legends, etc.) by 100-150% to ensure readability. We hope that the current versions of figures are sufficient for understanding.
Q2. The authors should explain why it was necessary to study LTR retrotransposons in two different plant species. What important information did the authors learn by using these two species together?
A2: As noted in the Introduction, our current understanding of TEs DNA methylation in plants is largely based on studies of species with relatively small genomes, such as Arabidopsis thaliana. However, research in large-genome plants, such as maize, has revealed substantial differences in both the distribution and maintenance of DNA methylation over LTR-RTEs. Therefore, we aimed to compare the methylation patterns of LTR-RTEs between species with large and small genomes. Indeed, our analysis revealed considerable genome-wide differences. At the same time, this comparative approach also showed that the Cme distribution pattern of individual LTR-RTE lineages can be remarkably conserved across species. We have clarified this point and revised Abstract accordingly.
Q3. Why LTR retrotransposon and not other transposons?
A3. The LTR-RTEs make up the majority of the sunflower’s mobilome composition (https://doi.org/10.1186/1471-2164-14-686, https://doi.org/10.1093/gbe/evv230, https://doi.org/10.1111/j.1365-313X.2012.05072.x) while other types of transposable elements are in neglectable quantities. To improve clarity we have modified the 4.7 section of Material and Methods (Transposable Elements’ Annotation).
Q4. Authors should include the Table with basic data of the nanopore sequencing for for each sample (total reads, reads after cleaning, GenBank accession numbers, etc).
A4. The table with the information about read number etc. has been added as a Supplementary file.
Q5. Authors should improve all legends for figures and tables – explain used abbreviations, samples.
A5. Corrected, we added more detailed descriptions for each figure.
Minor:
Q6. Line 523. “material, 1 ml of (Evrogen, Moscow, Russia) was” - 1 ml of what?
A6. Corrected.
Q7. Line 552. “4.4. Synthesis of ds-cDNA” – explain ds-cDNA abbreviation.
A7. Corrected.
Reviewer 3 Report
Comments and Suggestions for Authors
The study "Tiurin et al. Connecting Epigenetics..." addresses a relevant. Methylation plays an important role in regulating the transposition of mobile elements. The work was performed at a high level of modernity using nanopore sequencing. The results are relevant to the scope of the journal Plants. The authors obtained new data. No plagiarism was detected. The text of the article is clear. The illustrations are well presented. The following questions arose while reading the article:
- Line 4. Remove the period at the end of the title.
- The abstract should highlight the practical significance of the study. It is also advisable to add a sentence at the beginning outlining the relevance of this research area.
- Authors should avoid repeating terms in the title and keywords.
- Line 38. The range is clearly much wider. It is advisable to tone down the sentence, as these percentages vary greatly not only among different species and individuals, but also within the same organism.
- 59. de novo – italic
- 140. Asteraceae family – directly
- The article does not explain why the reverse transcriptase gene was used in the phylogenetic analysis. Also, neither the text nor the figure caption mentions that the retrotransposon reverse transcriptase gene was used in the analysis. This issue needs to be clarified.
- Figure 3A. There is no reference to this figure in the text.
- Figure 2A and Figure 3A. It is inappropriate to provide a phylogenetic tree in an article if nothing is clear from the figure. Apparently, this data should be presented in a different, more visual form.
- Figure 4, Figure 5. There is no clear explanation of the meaning of numbers 1, 2, and 3.
- 277. Two dots.
- The discussion clearly reflects the author's understanding of scientific novelty. However, there is no information on the practical significance of the results obtained.
- 523. What was added? Not stated.
- 572. Perhaps Sibenzyme, Novosibirsk was meant.
- The authors should more thoroughly check the entire article for errors.
The article may be accepted for publication after the authors have made the necessary edits.
Author Response
We thank the reviewer for the comments. Please find our point-by-point answers below. We believe these revisions have elevated the manuscript’s clarity and impact.
Q1. Line 4. Remove the period at the end of the title.
A1. Corrected.
Q2. The abstract should highlight the practical significance of the study. It is also advisable to add a sentence at the beginning outlining the relevance of this research area.
A2: Our study is primarily fundamental, yet the epigenetic regulation of LTR-RTEs plays a crucial role in plant genome evolution, diversity, and breeding potential. To address this comment, we added an opening sentence to the Abstract outlining the relevance of this research area:
“Transposable elements (TEs) are ubiquitous components of plant genomes that profoundly influence plant diversity, adaptation, and genome structure.”
Q3. Authors should avoid repeating terms in the title and keywords.
A3. We thank the reviewer for this comment, we added proper non-overlapping with title term keywords.
The revised version now reads:
“nanopore sequencing, sunflower, Arabidopsis, retrotransposons, DNA methylation profiling, epigenetics, comparative genomics”
Q4. Line 38. The range is clearly much wider. It is advisable to tone down the sentence, as these percentages vary greatly not only among different species and individuals, but also within the same organism.
A4. We thank the reviewer for this valuable comment.
The revised version now reads:
“Plant genomes exhibit substantial diversity in the TE content. For example, about ~21% of the Arabidopsis thaliana L. genome is occupied by transposons, whereas nearly 85% of the maize genome is composed of TEs sequences”
Q5. 59. de novo – italic
A5. Corrected.
Q6. Line 140. Asteraceae family – directly
A6. Corrected.
Q7. The article does not explain why the reverse transcriptase gene was used in the phylogenetic analysis. Also, neither the text nor the figure caption mentions that the retrotransposon reverse transcriptase gene was used in the analysis. This issue needs to be clarified.
A7: The reverse transcriptase (RT) domain is widely used for LTR-RTE phylogeny due to its high conservation as a protein-coding sequence (doi:10.1186/s13100-018-0144-1). We have added a corresponding description in the Materials and Methods section.
Q8. Figure 3A. There is no reference to this figure in the text.
A8. Corrected.
Q9. Figure 2A and Figure 3A. It is inappropriate to provide a phylogenetic tree in an article if nothing is clear from the figure. Apparently, this data should be presented in a different, more visual form.
A9. We agree and have updated the figures to present the phylogenetic trees in a simpler and clearer form, highlighting their relationship to the heat maps.
Q10. Figure 4, Figure 5. There is no clear explanation of the meaning of numbers 1, 2, and 3.
A10. A corrected version of figures with mention of cluster numbers was added.
Q11. Line 277. Two dots.
A11. Corrected.
A12: Our research is primarily fundamental, focusing on elucidating epigenetic mechanisms of LTR retrotransposons rather than applied outcomes. While direct practical applications remain speculative at this stage, we have added a brief statement to the Discussion:
“This finding also provides a foundation for understanding the mechanisms controlling this variation and may inform future development of novel TE-based mutagenesis strategies and epigenetic diversification approaches for crop improvement”
Q13. Line 523. What was added? Not stated.
A13. Corrected.
Q14. Line 572. Perhaps Sibenzyme, Novosibirsk was meant.
A14. Corrected.
Q15. The authors should more thoroughly check the entire article for errors.
A15: We have conducted a comprehensive review of the entire manuscript, systematically checking for grammatical, typographical, formatting, and scientific accuracy errors across all sections.
Round 2
Reviewer 2 Report
Comments and Suggestions for Authors
1) “Q4. Authors should include the Table with basic data of the nanopore sequencing for each sample (total reads, reads after cleaning, GenBank accession numbers, etc).
A4. The table with the information about read number etc. has been added as a Supplementary file.”
Is it Supplementary Table S1? There is very little data in this table and no accession numbers in the GeneBank. E.g. genomic DNA - where was the tissue taken for DNA extraction? Why are there few samples extracted from Arabidopsis? Why is the number of repetitions different in the samples? and much more
Comments on the Quality of English LanguageA small edit is needed
Author Response
We thank the reviewer for the additional comments. We have revised Table S1 and added the GenBank accession IDs for all datasets used in our study. Data for both replicates for sunflower and Arabidopsis have been deposited in GenBank. For both species, only leaf material was used for DNA sequencing. The Arabidopsis data were obtained from our previous study.
In addition to these corrections in Table S1, we carefully reviewed the entire manuscript and made minor improvements throughout the text. We also updated the Data Availability statement as follows:
The nanopore whole-genome sequencing (WGS) and RNA-seq data for Helianthus annuus L. generated in this study are available in the NCBI Sequence Read Archive (SRA) under BioProject accession PRJNA1394510. Data for Arabidopsis thaliana L. (wild-type Col-0 ecotype) is available in the BioProject accession PRJNA1263850.